# An enhancer screen identifies new suppressors of small-RNA-mediated epigenetic gene silencing

**Yukiko Shimada[1], Sarah H. Carl [1,2], Merle Skribbe[1,3], Valentin Flury [1,3], Tahsin Kuzdere[1,3], Georg Kempf [1], Marc Bühler [1,3]***

**1** Friedrich Miescher Institute for Biomedical Research, Basel, Switzerland, **2** Swiss Institute of Bioinformatics (SIB), Lausanne, Switzerland, **3** University of Basel, Basel, Switzerland

* marc.buehler@fmi.ch

## Abstract

Small non-protein coding RNAs are involved in pathways that control the genome at the level of chromatin. In *Schizosaccharomyces pombe*, small interfering RNAs (siRNAs) are required for the faithful propagation of heterochromatin that is found at peri-centromeric repeats. In contrast to repetitive DNA, protein-coding genes are refractory to siRNA-mediated heterochromatin formation, unless siRNAs are expressed in mutant cells. Here we report the identification of 20 novel mutant alleles that enable *de novo* formation of heterochromatin at a euchromatic protein-coding gene by using *trans*-acting siRNAs as triggers. For example, a single amino acid substitution in the pre-mRNA cleavage factor Yth1 enables siRNAs to trigger silent chromatin formation with unparalleled efficiency. Our results are consistent with a kinetic nascent transcript processing model for the inhibition of small-RNA-directed *de novo* formation of heterochromatin and lay a foundation for further mechanistic dissection of cellular activities that counteract epigenetic gene silencing.

**Data Availability Statement:** Small RNA and total RNA sequencing data have been deposited at the NCBI Gene Expression Omnibus (GEO) database and are accessible through GEO series number

## Author summary

Besides silencing gene expression at the post-transcriptional level, small RNAs mediate the formation of silent chromatin that is heritable across generations. Over the last two decades, fission yeast has been serving as an excellent model organism to elucidate the mechanism of small-RNA-mediated heterochromatin formation at repetitive DNA. More recently, work performed with fission yeast revealed the existence of cellular activities that prevent small RNAs from triggering the formation of heterochromatin outside repetitive DNA. With the current work we are expanding the list of factors involved in these counteracting mechanisms. Our results support a model in which small-RNA-directed epigenetic gene silencing is controlled by pre-mRNA cleavage and underscore the importance of the mRNA 3'end processing machinery in warranting gene expression. Because the list of experimentally determined alleles that allow small-RNA-mediated heterochromatin formation keeps expanding, we speculate that fission yeast's natural ecology may lead to the acquisition of silencing enabling genetic mutations as part of a biological bet-hedging

GSE173837. All other relevant data are within the manuscript and its Supporting Information files.

**Funding:** This work has received funding from the European Research Council (ERC) under the European Union's Horizon 2020 research and innovation programme (grant agreement no. 681213 - REpiReg) (M.B.) and the Novartis Research Foundation (M.B.). The funders had no role in study design, data collection and analysis, decision to publish, or preparation of the manuscript.

**Competing interests:** The authors have declared that no competing interests exist.

strategy. We therefore advocate for the inclusion of non-laboratory strains in future research that aims at understanding the physiological relevance of small-RNA-mediated epigenetic gene silencing.

## Introduction

Small RNAs are the common denominator of various RNA silencing pathways that regulate gene expression and protect the genome against mobile repetitive DNA sequences, retroelements, and transposons [1–3]. They function as specificity factors by guiding Argonaute protein-containing silencing complexes to their respective targets via base-pairing interactions [4, 5]. In the fission yeast *Schizosaccharomyces pombe*, endogenous small interfering RNAs (siRNAs) are indispensable for the maintenance of centromeric heterochromatin [6]. They originate from within centromeric heterochromatin and target the Argonaute-containing (Ago1) RNA-Induced Silencing Complex (RITS) *in cis* to nascent heterochromatic transcripts that are emanating from RNA polymerase II (RNA Pol II) transcribing the underlying repetitive DNA [7, 8]. Besides binding to nascent RNAs, RITS also binds to methylated histone H3 lysine 9 (H3K9me) through its chromodomain-containing subunit Chp1 [9, 10]. Constituting a positive feedback loop, the RITS complex recruits H3K9 methylation and RNA-dependent RNA polymerase activities to the locus it associates with [9, 11–13]. Dicer-mediated (Dcr1) processing of the resulting double-stranded RNAs leads to amplification of the siRNA pool and thereby reinforcement of the positive feedback loop [14].

Whereas siRNAs originating from heterochromatic repeats function well *in cis* to sustain H3K9 methylation, they do not act *in trans* to mediate *de novo* formation of heterochromatin at complementary protein-coding genes outside centromeric repeats in wild-type cells [15]. Similarly, synthetic siRNAs produced from RNA-hairpins are not sufficient to stably silence homologous protein-coding genes through the assembly of heterochromatin [16–19]. However, siRNAs have been shown to become potent mediators of RNAi-mediated epigenetic gene silencing in *S. pombe* cells that are mutant for $mlo3^+$, $dss1^+$, $mst2^+$, or genes encoding subunits of the Paf1 complex (Paf1C) [15, 18, 20, 21]. Indicating potential evolutionary conservation, Paf1C also opposes PIWI/piRNA-directed silencing in *Drosophila melanogaster* [22].

Current understanding of the mechanisms that counteract small-RNA-mediated epigenetic gene silencing is in its infancy. Furthermore, rates at which silencing is initiated or maintained vary substantially between the different enabling mutations identified so far. For example, initiation of gene silencing was reported to occur in approximately 0.5–2% of *mlo3Δ* cells [15]. Silencing in Paf1C mutants is initiated in up to 20% of cells and is boosted to more than 80% upon additional deletion of the $mst2^+$ gene [18, 20]. Although initiation of heterochromatin assembly in *mlo3Δ* and Paf1C mutant cells is not efficient, the silent state is stably maintained once established, even when cells undergo meiosis. That is, Mendelian segregation of the silent allele was observed irrespective of the primary siRNA trigger and enabling $mlo3^+$ deletion in one study [15]. Inheritance of the repressed state induced by hairpin-derived siRNAs remains dependent on the enabling mutations in Paf1C but ensues independently of the hairpin trigger as well [15, 18, 23]. Here, inheritance patterns of the silencing phenotype violate Mendel's laws, reminiscent of the paramutation phenomenon [18, 23, 24].

Whereas the role of the RNA export factor Mlo3 in counteracting small-RNA-mediated epigenetic gene silencing remains enigmatic, two non-mutually exclusive models have been put forward for Paf1C [25]. The first model suggests that dilution of K9 methylated H3 is lowered by the reduced histone H3 exchange rates in Paf1C mutants, stabilizing the H3K9

methylation state and hence the aforementioned positive feedback loop [26]. The second model suggests that mutations in Paf1C reduce the kinetics of nascent transcript release from chromatin, allowing sufficient time for RITS to recruit H3K9 methylation and RNA-dependent RNA polymerase activities that are necessary to initiate and propagate the positive feedback loop [18]. This model is supported by observations that alterations in the polyadenylation signal (PAS) of a target pre-mRNA enable siRNA-directed H3K9 methylation [27]. Yet, mutations in the pre-mRNA cleavage and polyadenylation machinery that would impair nascent transcript cleavage and hence potentially enable small-RNA-directed *de novo* formation of heterochromatin formation have not been identified. Thus, additional evidence supporting the second model is wanted.

In this study we have combined chemical mutagenesis with whole-genome sequencing in a sensitized reporter strain to obtain a more comprehensive list of putative suppressors of small-RNA-mediated epigenetic gene silencing. This revealed more than 20 novel silencing-enabling mutations in genes that are associated with RNA processing, regulation of transcription, or post-translational protein modification. Focusing on factors involved in pre-mRNA cleavage and polyadenylation, we show that single amino acid substitutions in Yth1, which is responsible for PAS recognition, lead to nearly 100% effective *de novo* formation of silent heterochromatin. Our work provides further support for a kinetic model for the inhibition of small-RNA directed *de novo* formation of heterochromatin and demonstrates that epigenetic gene silencing can be enabled by the acquisition of a plethora of mutant alleles in fission yeast.

## Results

### An enhancer screen identifies 20 novel mutant alleles that enable small-RNA-mediated epigenetic gene silencing

To identify novel factors that suppress the susceptibility of protein-coding genes for epigenetic silencing via siRNAs that are acting *in trans*, we employed an *ade6⁺*-based reporter system and whole-genome sequencing pipeline of a previous screen that had revealed Paf1C as a potent inhibitor of siRNA-directed heterochromatin formation [18]. *ade6⁺* is a suitable reporter because Ade6-deficient cells form red colonies on limiting adenine indicator plates, whereas *ade6⁺* expressing cells appear white. This allows simple assessment and quantification of the initiation, maintenance, and inheritance of siRNA-mediated silencing [28].

Because deletion of the *mst2⁺* gene substantially increases the rate at which silencing is established *de novo* in Paf1C mutant cells [20], we created a sensitized reporter strain in which the *mst2⁺* gene was deleted, and a RNA hairpin (ade6-hp) complementary to 250 nt of the *trp1⁺*::*ade6⁺* reporter was expressed from the *nmt1+* locus on chromosome I (Fig 1A). In the absence of additional enabling mutations, the siRNAs generated from the ade6-hp do not stably silence the complementary *trp1⁺*::*ade6⁺* reporter gene *in trans* in a *mst2ᐞ* background [20] (Fig 1A). To screen for mutants that would enable *ade6* siRNAs to establish and maintain robust *trp1⁺*::*ade6⁺* silencing, we mutagenized our sensitized reporter strain with ethylmethansulfonate (EMS). This was followed by several selection and triaging steps before the introduced mutations were finally mapped by whole-genome sequencing (Fig 1B).

Of roughly 280'000 EMS treated single cell derived colonies, about 700 colonies showed red or red/white variegating phenotypes. To select against loss-of-function mutations in the adenine biosynthesis pathway, these colonies were grown in the absence of adenine. Clones that were still able to grow were subsequently shifted to 37˚C. At this temperature, the repressed state of a heterochromatinized *ade6⁺* reporter gene is reversed, i.e. white instead of red colonies are formed. At this step, we ended up with 103 colonies in which the *trp1⁺*::*ade6⁺* reporter gene was silenced epigenetically (showing white and red phenotypes at 37˚C and 30˚C,

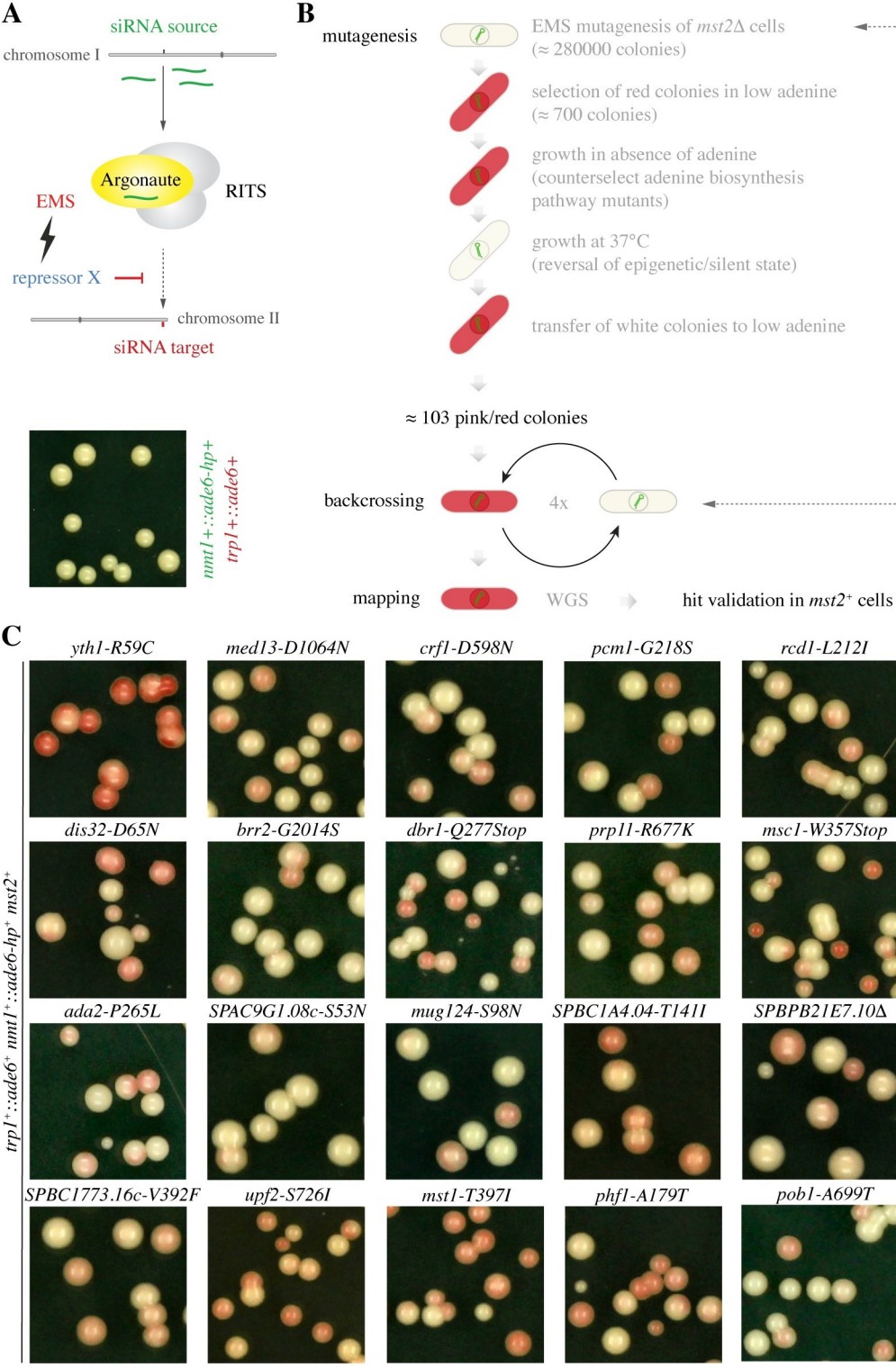

**Fig 1. Enhancer screen to identify mutant alleles that enable small-RNA-mediated epigenetic gene silencing. (A)** Schematic representation of the tester strains ($mst2^+$ or $mst2^\Delta$) used in this study. Primary $ade6$ siRNAs are produced from the $nmt1^+$ locus on chromosome I (green). They can only induce silencing of the $ade6^+$ reporter gene on chromosome II, if the tester strain acquires an enabling mutation (white color of the colonies shown indicates expression of the $ade6^+$ reporter gene despite the presence of $ade6$ siRNAs in these cells). **(B)** Workflow of the EMS mutagenesis screen. For the initial screening, a $mst2^\Delta$ tester strain was used. Final hit validation was performed in

*mst2*⁺ cells. (**C**) Representative images of validated point-mutations that are sufficient to enable RNAi-directed *ade6*⁺ silencing, which is indicated by the red color. Colonies showing a silencing phenotype on YE-NAT (100ug/ml nourseothricin) plates were selected and spread on YE plates to monitor stability of *ade6*⁺ repression. Cells are *mst2*⁺ but harbor mutations in individual genes as indicated.

respectively). After these were backcrossed four times, mutations that segregated with an *ade6*⁺ repression phenotype were mapped by whole-genome next-generation sequencing (S1 Table). Validating our screen, four clones had acquired mutations in subunits of the Paf1C complex. In another clone we found a nonsense mutation in the *res2*⁺ gene, which we had previously shown to display a weak silencing phenotype when deleted (Table 1) [18]. Thus, our screen reliably identifies mutants that enable small-RNA-mediated epigenetic gene silencing, even if initiation rates are poor.

To validate the mapped sequence alterations (S1 Table) as the causative mutations, and to test if they could also function independently of impaired Mst2 activity, we reconstituted the candidate point mutations in our original tester strain (*mst2*⁺) [18]. This revealed 20 novel silencing-enabling point mutations, which reliably recapitulated the *ade6*⁺ silencing phenotype (Fig 1C). Consistent with small-RNA-mediated epigenetic silencing responses, the *ade6*⁺ repression phenotypes were reversible and depended on a functional *dcr1*⁺ allele in all 20 strains (S1 and S2 Figs). Eight of these novel enabling mutations were found in genes associated with RNA processing, four in genes encoding regulators of transcription, and three in genes that have been implicated in post-translational modification of histones. Another five mutations were found either in genes of unknown function or in genes related to lipids (Table 2). In conclusion, our enhancer screen has identified 20 novel high confidence alleles that enable siRNAs to induce gene silencing *in trans*.

## Arginine to cysteine substitution in the Yth1 cleavage/polyadenylation factor enables efficient initiation of heterochromatin-mediated gene silencing

Among all mutants tested, cells with an arginine to cysteine mutation at position 59 in the *yth1*⁺ gene (*yth1-R59C*) displayed the strongest silencing phenotype (Figs 1C, 2A and 2B). The Yth1 protein is a subunit of the cleavage and polyadenylation factor complex (CPF) and is responsible for the recognition of the AAUAAA polyadenylation signal in pre-mRNAs [29]. This is interesting because a previous study highlighted the importance of the PAS in preventing the formation of heterochromatin [27].

Stability of a heterochromatin-mediated silencing phenotype depends on the rate at which heterochromatin is established, or on the robustness of the mechanisms that preserve the silent chromatin state through mitosis, or both. For example, in Paf1C mutant cells, silencing is established rather inefficiently, but it is very stably propagated through subsequent cell divisions [18, 20](Fig 2C and 2D). In *yth1-R59C* cells, we observed an initiation rate of silencing that was close to 100% (Fig 2C). Silencing was maintained stably but appeared more

**Table 1. Anticipated genes with mutations mapped by whole-genome sequencing.**

| systematic ID | protein | mutation |
|---|---|---|
| SPAC664.03 | Paf1, Paf1 complex | G102S |
| SPAC664.03 | Paf1, Paf1 complex | Q170Stop |
| SPBC651.09c | Prf1, Paf1 complex | E435K |
| SPBC651.09c | Prf1, Paf1 complex | A439T |
| SPAC22F3.09c | Res2 | Q41Stop |

**Table 2. Newly identified and validated enabling mutations.**

| systematic ID | protein | complex | mutation | biological process | dcr1Δ |
|---|---|---|---|---|---|
| SPAC227.08c | Yth1 | CPF | R59C | polyadenylation signal recognition | √ |
| SPAC29B12.06c | Rcd1 | CCR4-NOT | L212I | poly(A) tail shortening | √ |
| SPCC330.10 | Pcm1 | Capping complex | G218S | mRNA capping | √ |
| SPCC10H11.01 | Prp11 | U2 prespliceosome | R677K | mRNA splicing | √ |
| SPAC17A5.02c | Dbr1 | spliceosome | Q277Stop | mRNA splicing, RNA lariat debranching | √ |
| SPAC9.03c | Brr2 | U5 snRNP | G2014S | mRNA splicing | √ |
| SPAC2C4.07c | Dis32 | . . . | D65N | 3'-5' RNA degradation | √ |
| SPAC19A8.08 | Upf2 | . . . | S726I | nonsense-mediated mRNA decay | √ |
| SPAC589.02c | Med13 | Mediator | D1064N | regulation of transcription | √ |
| SPAC22H10.11c | Crf1 | . . . | D598N | regulation of transcription, TOR signaling | √ |
| SPBC1773.16c | . . . | . . . | V392F | regulation of transcription, DNA binding | √ |
| SPCC24B10.08c | Ada2 | SAGA | P265L | regulation of transcription, stress response | √ |
| SPAC343.11c | Msc1 | Swr1 complex | W357Stop | H3K36 demethylation, histone exchange | √ |
| SPAC637.12c | Mst1 | NuA4 | T397I | histone acetylation | √ |
| SPCC4G3.07c | Phf1 | Lsd1/2 complex | A179T | histone H3K9 demethylation | √ |
| SPBC19C2.06c | Mug124 | . . . | S98N | S. pombe specific protein, uncharacterized | √ |
| SPBC1A4.04 | . . . | . . . | T141I | S. pombe specific protein, uncharacterized | √ |
| SPBPB21E7.10 | . . . | . . . | Q2Stop | S. pombe specific protein, uncharacterized | √ |
| SPAC9G1.08c | . . . | . . . | S53N | protein depalmitoylation | √ |
| SPBC1289.04c | Pob1 | . . . | A699T | lipid binding, regulation of exocytosis | √ |

variegating than the silencing phenotype observed in Paf1C mutants (Fig 2A and 2D). Furthermore, we also observed silencing in *yth1-R59C* cells that express synthetic ura4-hp siRNAs instead of ade6-hp siRNAs, and a *trp1⁺::ura4⁺* instead of a *trp1⁺::ade6⁺* reporter (S3A Fig). Thus, we conclude that the *yth1-R59C* allele enables *trans*-acting siRNAs to effectively initiate the formation of heterochromatin at their target locus.

The silencing phenotypes described above, together with its temperature-sensitivity, strongly imply RNAi-mediated heterochromatin formation at the target locus. To formally demonstrate this, we assessed initiation of silencing of the *trp1⁺::ade6⁺* reporter gene in *yth1-R59C* cells lacking either a functional *dcr1⁺* gene or the primary siRNA producing *nmt1⁺::ade6-hp⁺* locus. We observed silencing in neither of these strains (Fig 2B), demonstrating the necessity of siRNA biogenesis. To confirm the formation of a heterochromatic structure at the *trp1⁺::ade6⁺* locus, we performed chromatin immunoprecipitation (ChIP) experiments using an antibody specifically recognizing di-methylated lysine 9 on the N-terminal tail of histone H3 (H3K9me2). As predicted, the H3K9me2 mark was significantly enriched in *yth1-R59C* cells. *yth1⁺* cells were not different from cells lacking Clr4, which is the sole H3K9 methyltransferase in *S. pombe* (Fig 2E). Finally, assembly of heterochromatin at the *trp1⁺::ade6⁺* reporter gene was accompanied by the production of secondary *ade6⁺* siRNAs that are not encoded in the ade6-hp (Fig 2F).

These results demonstrate that replacing arginine at position 59 of Yth1 with a cysteine does not critically affect expression of the *trp1⁺::ade6⁺* reporter gene (Fig 2B). However, it enables highly efficient initiation of heterochromatin-mediated gene silencing upon expression of primary siRNAs that are complementary to the *ade6⁺* pre-mRNA (S3B and S3C Fig).

## PAS recognition controls small-RNA-mediated epigenetic gene silencing

CPF is a large multisubunit protein complex possessing ATP-polynucleotide adenylyltransferase, phosphatase, and nuclease activities that are required for the cleavage and polyadenylation

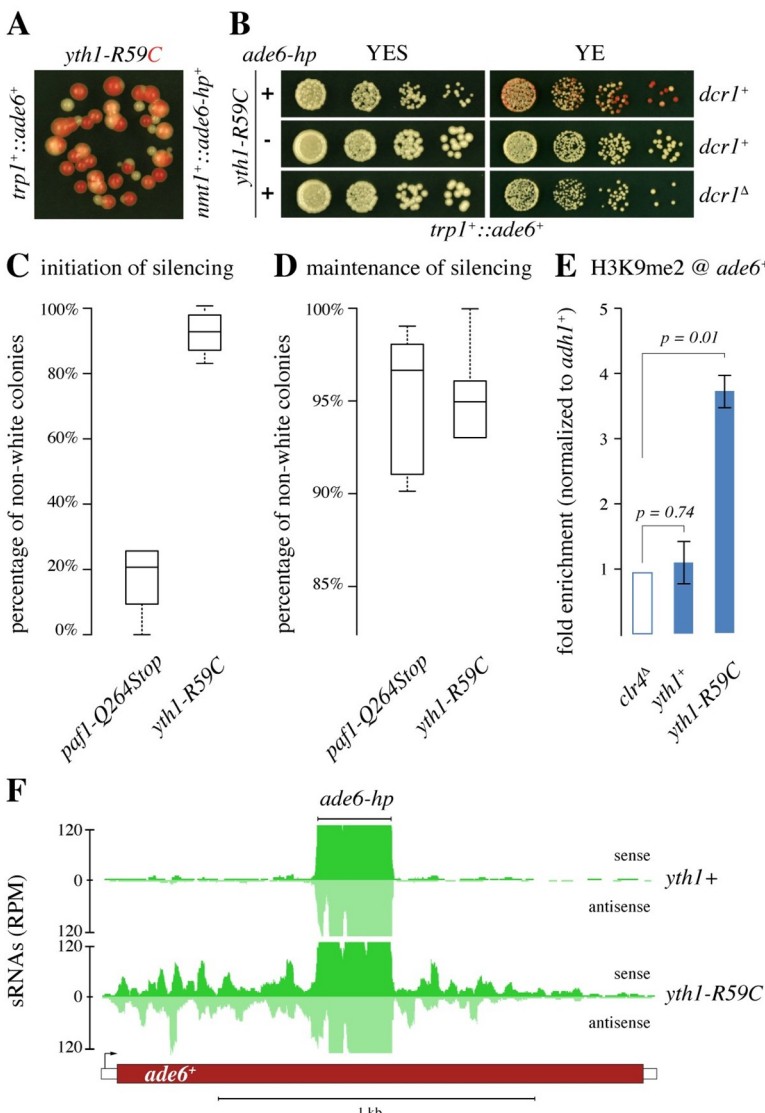

**Fig 2. Arginine to cysteine substitution in Yth1 enables small-RNA-directed heterochromatin assembly. (A and B)** Silencing assays showing the *ade6+* silencing phenotype in *yth1-R59C* mutant cells. Silencing in individual colonies is typically variegating (A), which is lost in cells lacking either Dicer (*dcr1ᐞ*) or the ade6 RNA hairpin (*ade6-hp*) (B). **(C and D)** Comparison of initiation (C) and maintenance (D) of *ade6+* silencing in *paf1-Q264Stop* and *yth1-R59C* cells (*trp1+::ade6+*, *nmt1+::ade6-hp+*). Multiple individual originator colonies (white or red in C or D, respectively) were spread to single cell density on YE plates to assess initiation/maintenance of the silencing phenotype: n = 6 for *paf1-Q264Stop* white originator (C, total counted number of colonies = 2547), n = 6 for *paf1-Q264Stop* red originator (D, total counted number of colonies = 3016), n = 12 for *yth1-R59C* white originator (C, total counted number of colonies = 4063), n = 9 for *yth1-R59C* red originator (D, total counted number of colonies = 9468). **(E)** ChIP analysis of H3K9me2 in the strains indicated (*trp1+::ade6+*, *nmt1+::ade6-hp+*). Fold enrichments were normalized to *adh1+* and are shown relative to background levels measured in *clr4ᐞ* cells. Error bars indicate standard deviation, n = 3 independent biological replicates, p-values were calculated with a two-tailed Student's t-test. The *ade6+* primer pairs do not discriminate between endogenous *ade6-704* and *trp1+::ade6+* genes. **(F)** Small RNA sequencing was performed with *yth1+* and *yth1-R59C* cells to assess secondary *ade6+* siRNA production. Read counts were normalized to library size. The part of *ade6+* that is complementary to the primary siRNAs encoded by the *ade6-hp* is denoted.

of pre-mRNA transcripts [29]. Yth1 is part of the poly(A)polymerase module of CPF and is essential for cellular viability [30]. Because 3' end processing of pre-mRNAs has previously been implicated in the control of RNA silencing pathways in yeast, flies, and plants [18, 27,

31–35], the identification of *yth1-R59C* as a silencing-enabling allele is appealing. Because our EMS screen revealed only one enabling mutation residing in Yth1, we decided to perform a directed evolution experiment to select additional putative *yth1* alleles that would promote small-RNA-mediated epigenetic gene silencing.

We cloned the *yth1*+ gene in a *hph*+-marked plasmid (p-*yth1*+/*hph*+), which we subsequently propagated in the *E. coli* XL1-Red mutator strain to produce a randomly mutagenized plasmid library (p-*yth1**/*hph*+)[36]. This library was then transformed into our *S. pombe* tester strain (*mst2*+) [18], in which we deleted the endogenous *yth1*+ gene, to screen for silencing-enabling Yth1 mutants. Because *yth1*$^\Delta$ cells are not viable, this modified tester strain (*mst2*+) was rescued with a *ura4*+-marked *yth1*+ expression plasmid (p-*yth1*+/*ura4*+). Upon transformation with the mutant p-*yth1**/*hph*+ library, loss of the p-*yth1*+/ura4+ rescue plasmid was forced by growth on medium containing 5-fluoroorotic acid, which is toxic to *ura4*+ expressing cells (Fig 3A). Thus, complete loss of function Yth1 mutants were counter-selected during this step of the experiment. To find small-RNA-mediated epigenetic gene silencing-enabling Yth1 mutants, we grew the p-*yth1**/*hph*+ expressing tester strain (*mst2*+) on low adenine plates and isolated single colonies that displayed an *ade6*+ silencing phenotype (Fig 3B). Sanger sequencing of the recovered p-*yth1**/*hph*+ plasmids revealed seven residues that were mutated: 2x R59C, 1xK55N, 2x E73K, 1x Y74H, 4x C91Y, 1x C91R, and 1x Y99H (Fig 3C). Because the R59C mutation was found again twice and C91 was mutated five times, this screen might have reached saturation.

Because protein structures of *S. pombe* CPF have not yet been determined, we selected the structure of human CPSF-30 (homolog of *S. pombe* Yth1) in complex with CPSF-160 (homolog of *S. pombe* Cft1), WDR33 (homolog of *S. pombe* Pfs2), and PAS RNA as a homology model to infer the functional consequences of the enabling mutations that we have identified (PDB ID 6DNH) [37]. This revealed that mutations in E73 and Y74 are likely to disturb the Yth1-Cft1 interaction and the C91R mutation the Yth1-Pfs2 interaction, whereas the other mutations are predicted to weaken the interaction with the PAS in the pre-mRNA (Fig 3D). Interestingly, K55, R59, C91, and Y99 contribute to the binding of Yth1 to the adenosine at position 4 ($A^4$) of the PAS, suggesting that an adenosine at position 4 is critical for the prevention of silencing. Indeed, mutating $A^4$ in the *trp1*+::*ade6*+ reporter to either U, C, or G enabled siRNAs to initiate silencing (Figs 3E and S3B). Further supporting the importance of PAS recognition, we observed siRNA-dependent *trp1*+::*ade6*+ reporter silencing upon mutation of F115 in Pfs2 (Figs 3F and S3D), which stacks with $A^6$ of the PAS (Fig 3D).

Interestingly, initiation frequency of the silencing response was lower in *pfs2-F115H* cells than observed with $A^4$ PAS mutants (Fig 3E and 3G). Yet, once established the silent state was remarkably stably maintained (Fig 3G). We note that residue C91 is part of the second CCCH zinc finger motif in Yth1. Correct folding of this zinc finger is important for binding $A^4$/$A^5$ in the PAS as well as for the interaction of Yth1 with Pfs2 (Fig 3D). Thus, Yth1-C91 mutations are likely to stimulate both initiation and maintenance of silencing, providing a potential explanation why we have mapped C91 five times in this screen.

## Deletion of non-essential subunits of the phosphatase module of the cleavage and polyadenylation factor complex enables small-RNA-mediated epigenetic gene silencing

The foregoing results implicate the fission yeast CPF in the control of small-RNA-mediated epigenetic gene silencing. Because we have mapped enabling mutations in the poly(A) polymerase-module of CPF only, we asked whether mutations in the other modules would similarly enable silencing. Unfortunately, the genes encoding subunits of the nuclease module are

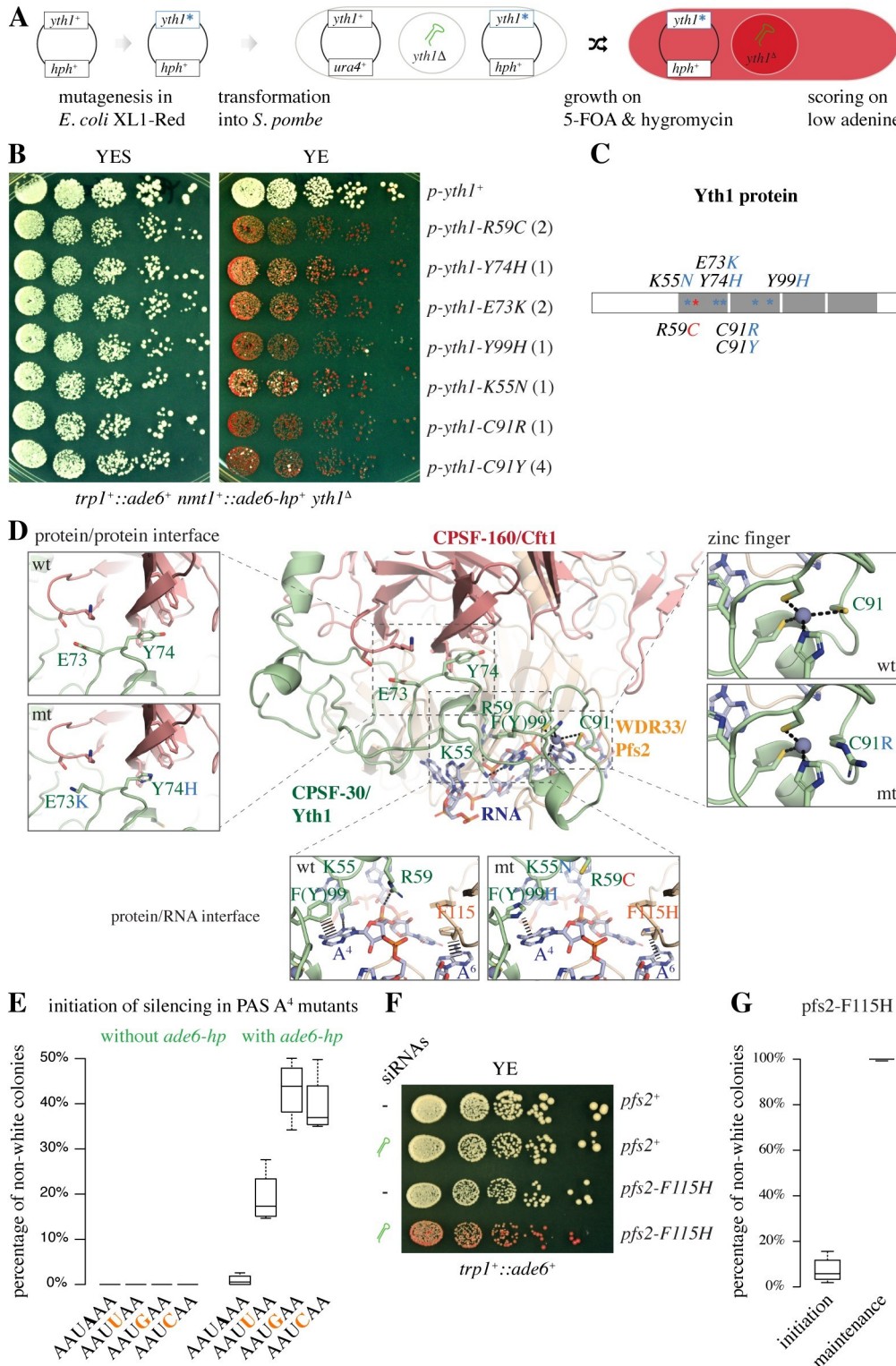

**Fig 3. Weakening CPF/AAUAAA RNA interactions enables small-RNA-mediated epigenetic gene silencing. (A)** Workflow of the *yth1+* random mutagenesis screen. The mutant *yth1\** plasmid library was transformed into the tester strain (*mst2+*) lacking the endogenous *yth1+* gene. The *ade6-hp* construct (primary siRNA source) is indicated in green. **(B)** *yth1* mutants that display an *ade6+* silencing phenotype on YE plates. The numbers in brackets denote how many times the respective mutation was recovered in the screen. **(C)** Domain organization of the Yth1 protein. Grey

boxes indicate the CCCH zinc finger domains. Asterisks denote the recovered mutations. (**D**) Modelling of point-mutations using PyMOL. The structure of human CPSF-30 in complex with CPSF-160, WDR33 and PAS RNA (PDB ID 6DNH) was selected as homology model for the *S. pombe* complex. Insets show wild-type residues compared to the modelled point mutations. The numbering corresponds to *S. pombe* Yth1. Zinc ions are shown as spheres. Putative π-stacking interactions are indicated by dashes with varying broadness based on estimated interaction strength. (**E**) Initiation frequency of *ade6*⁺silencing in cells harboring mutations at the 4ᵗʰ position in the PAS of the *ade6*⁺ reporter gene. At least eight individual white originator colonies per indicated genotype were spread to single cell density on YE plates. Total number of colonies counted: 1734 (AAUAAA without siRNAs), 1605 (AAUUAA without siRNAs), 1324 (AAUGAA without siRNAs), 1312 (AAUCAA without siRNAs), 1957 (AAUAAA with siRNAs), 1790 (AAUUAA with siRNAs), 2314 (AAUGAA with siRNAs), 2144 (AAUCAA without siRNAs). Primary siRNAs are encoded by the *ade6-hp* construct. (**F**) Silencing assay demonstrating siRNA-directed *ade6*⁺ silencing in *pfs2-F115H* cells specifically. The *ade6-hp* construct (primary siRNA source) is indicated in green. (**G**) Initiation and maintenance of *ade6*⁺ silencing in *pfs2-F115H* cells. Four individual originator colonies (white for initiation, red for maintenance) were spread to single cell density on YE plates. 361 and 665 colonies were counted to determine initiation and maintenance frequencies, respectively.

essential for viability [30], preventing us from testing those. However, we were able to delete the *dis2*⁺, *ppn1*⁺, *swd22*⁺, and *ssu72*⁺ genes, which encode subunits of the phosphatase module (Fig 4A). Like the poly(A) polymerase module mutants, the four phosphatase module knock-out strains enabled small-RNA-mediated gene silencing (Figs 4B and 4C and S4A–S4C). Interestingly, Swd22 and Ssu72 have recently been shown to be required for RNAi-independent assembly of facultative heterochromatin [32]. Thus, the phosphatase module of CPF may have opposing roles depending on the pathway that leads to H3K9 methylation.

A remarkable feature of CPF mutants that we have investigated in this study is that they by and large phenocopy wild-type cells, i.e. neither growth nor global gene expression is largely affected (S5 Fig). For example, we did not observe any major differences in steady-state *ade6*⁺ mRNA levels in CPF or PAS mutant cells in the absence of *ade6* siRNAs (Figs 5A and 5B and S5A). Also, polyadenylation of *ade6*⁺ mRNA in wild-type and mutant cells was indistinguishable in our assay (Fig 5B). However, we observed compromised cleavage of the *ade6*⁺ pre-mRNA, which was prominent in many of the mutants investigated (Fig 5A upper panel and 5C). This strongly implies reduced kinetics of the 3' end processing reaction.

In conclusion, our results are consistent with previous works that have implicated pre-mRNA processing factors in small-RNA-mediated silencing responses [15, 18, 31]. Our detailed analyses of *S. pombe* CPF mutants reinforce the importance of an efficient 3' end processing reaction to avoid an unwanted gene silencing response.

## Discussion

In this study we have identified novel mutant alleles that make *S. pombe* susceptible for RNAi-mediated *de novo* assembly of silent chromatin. Though we have focussed on a functional dissection of mutations in the pre-mRNA cleavage and polyadenylation machinery in this study (Fig 4A), it will be equally exciting to dissect the role of the other RNA processing factors that our screen has revealed. Likewise, further investigating the many alleles linked to chromatin biology or transcription regulation promises to further improve our understanding of the intricate mechanisms that keep RNA-mediated epigenetic processes in check (Table 2).

Our work on CPF presented here is consistent with our previously proposed kinetic model for the inhibition of *de novo* formation of heterochromatin that is mediated by *trans*-acting primary siRNAs. In this model, the rate at which the nascent transcript is released from the DNA template is predicted to constitute a rate limiting step for the initiation of heterochromatin assembly and eventually gene silencing [18]. We find it striking that *ade6*⁺ pre-mRNA cleavage in *yth1-R59C* cells is similarly affected as in cells harbouring *ade6*⁺ genes with A⁴ mutated PASs (Fig 4B and 4C). As predicted by the kinetic model, mutations that we have

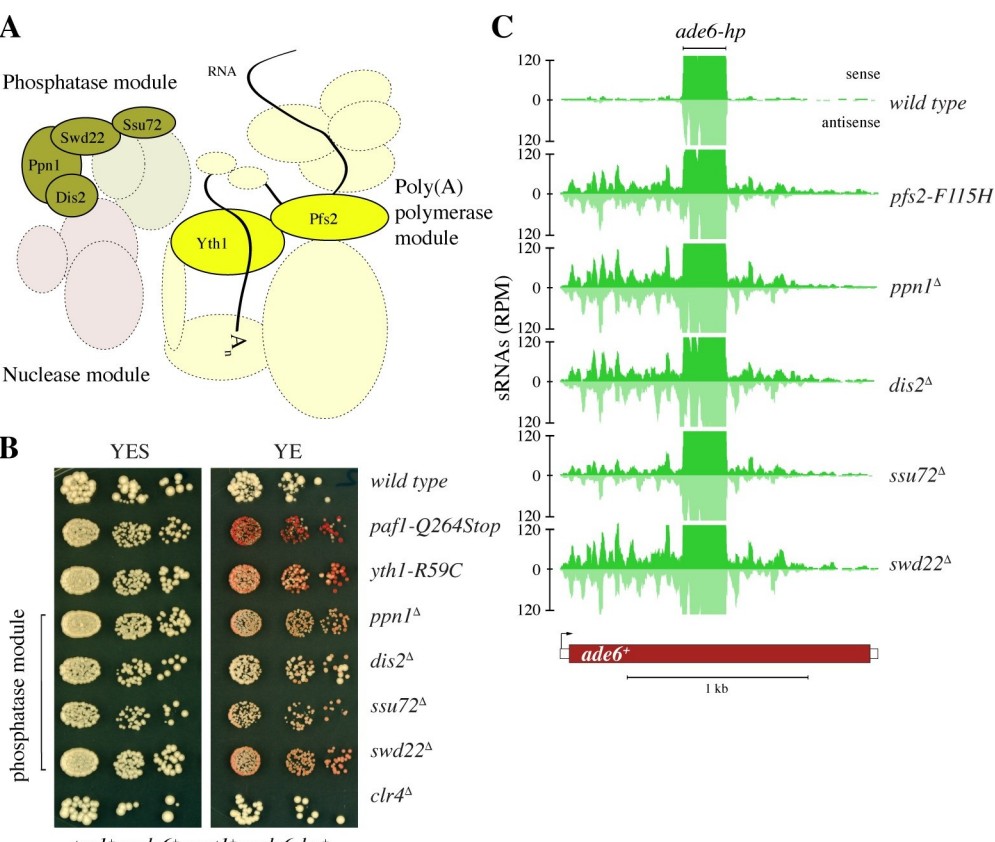

**Fig 4. Mutations in the phosphatase module of CPF enable small-RNA-mediated epigenetic gene silencing. (A)** Phosphatase, nuclease, and poly(A) polymerase modules of CPF, adapted from a model of budding yeast CPF [29]. Subunits found in this study to enable siRNA-mediated gene silencing when mutated are indicated by filled circles and their protein names. **(B)** Silencing assay showing the degree of *ade6⁺* silencing in cells with an impaired phosphatase module of CPF. Ssu72 and Dis2 are active phosphatases. See also S4 Fig. **(C)** Small RNA sequencing was performed with the strains indicated to assess secondary *ade6⁺* siRNA production. Read counts were normalized to library size. The part of *ade6⁺* that is complementary to the primary siRNAs encoded by the *ade6-hp* is denoted.

mapped in CPF are thus likely to result in decelerated cleavage and release of the nascent transcript from the site of transcription, opening up a window of opportunity for the siRNA-guided RITS complex to base-pair with pre-mRNA and recruit the histone methylation machinery. Such a model nicely explains why the initiation rates that we have observed in *yth1-R59C* cells are so remarkably high (Fig 2C).

Although we could not investigate factors of the CPF nuclease module, we deleted four genes that are encoding subunits of the phosphatase module. While to seemingly various degrees, all four mutants enabled silencing. This is intriguing because Ssu72 and Dis2 are active phosphatases, letting us to speculate that inhibition of RNAi-mediated heterochromatin formation could be regulated by kinase signalling pathways. This is of particular interest in light of a recent report that described the isolation of heterochromatin-dependent epimutants that are resistant to caffeine, which is abolished in RNAi mutants [38]. A similar phenomenon had been described earlier in *Mucor circinelloides*, which can cause deadly fungal infections in humans. Similar to *S. pombe* responding to low doses of caffeine, *M. circinelloides* can become resistant against antifungal drugs through RNAi-mediated epigenetic gene silencing [39, 40]. It is tempting to speculate that sensing of toxic substances in the environment could be

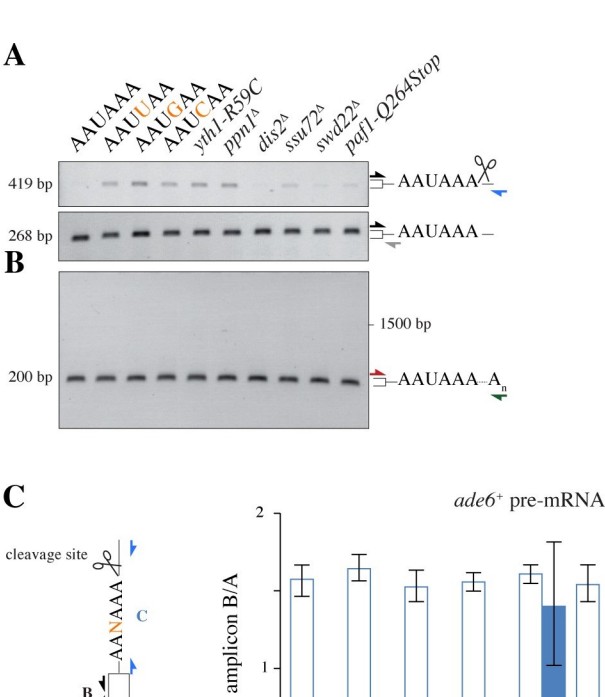

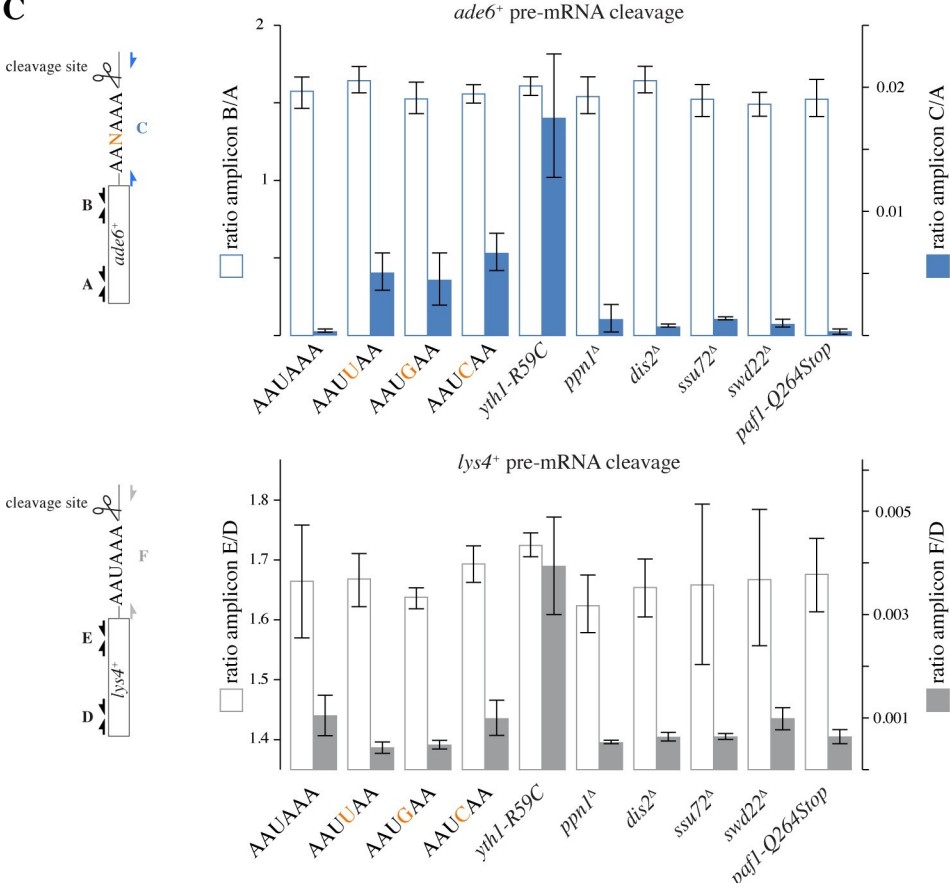

**Fig 5. Silencing enabling mutations in the PAS of a siRNA target pre-mRNA or the poly(A) polymerase module, but not the phosphatase module of CPF, affect pre-mRNA cleavage. (A)** Qualitative RT-PCR with primer pairs amplifying amplicons before (lower panel) and across (upper panel) the PAS and major cleavage site of the *ade6*+ pre-mRNA. See S4 Table for primer sequences. **(B)** RT-PCR amplifying polyadenylated *ade6*+ mRNA using an anchored oligodT primer (green arrow). **(C)** Quantitative RT-PCR with primer pairs binding to the ORF (amplicons A and B) or up- and downstream of the cleavage site (amplicon C) of the *ade6*+ pre-mRNA (upper panel), or primer pairs binding to the ORF (amplicons D and E) or up- and downstream of the cleavage site (amplicon F) of the *lys4*+ pre-mRNA (lower panel). Positions of the primer pairs are shown in the schematics on the left (not drawn to scale). Note that the PAS mutations indicated in the lower panel refer to the *ade6*+ PAS. The *lys4*+ PAS was not mutated in these samples. Error bars indicate standard deviation, n = 3 or 4 independent biological replicates for *lys4*+ or *ade6*+, respectively. (A—C) cDNA was prepared from RNA that was isolated from cells with the indicated genotype. These cells did not express *ade6*+ siRNAs.

signalled to the RNAi-inhibiting modules that we have identified in our works. Potential modulation of such signalling by the CPF phosphatases is an attractive hypothesis that will be worthwhile further investigation.

Although regulation of RNAi-mediated epigenetic gene silencing through direct environmental sensing is appealing, we do not exclude the possibility that RNAi-mediated *de novo* formation of heterochromatin on protein-coding genes strictly depends on the acquisition of an enabling mutation. In this model, an epigenetic gene silencing response would always be preceded by a genetic change. This is supported by the many different enabling genetic mutations that we and others have identified so far [15, 18, 20], and by our unsuccessful efforts to trigger small-RNA-mediated gene silencing under various environmental conditions in wild-type cells. Thus, we urge the community to consider the possibility that acquisition of a genetic mutation had preceded the establishment of the observed siRNA-triggered epigenetic silencing phenotype, especially when "on" and "off" expression states segregate with a 2:2 Mendelian ratio in seemingly wild-type cells [15]. As we have already discussed elsewhere, prior acquisition of RNAi-enabling genetic mutations could also explain why virulent isolates of *M. circinelloides* have an enhanced ability to develop drug resistance through epimutations [25].

The latter model would be fully consistent with the concept of biological bet-hedging, as enabling stochastic RNAi-mediated epigenetic silencing might help the microbe to adapt to an ever-changing environment [25]. Therefore, it could be advantageous for a yeast to hedge its bets by acquiring an enabling mutation in case its environment keeps changing. Because this comes along with decreased fitness in stable conditions, such mutations would be expected to disappear in a laboratory strain. This may explain why *S. pombe* cells that we are growing in our labs are refractory to RNAi-mediated gene silencing.

## Materials and methods

### Yeast strains

*S. pombe* strains were generated following a PCR-based protocol [41] or by standard mating and sporulation. For a list of strains generated in this study see S2 Table.

As a general procedure to validate the newly identified enabling alleles, the mapped mutations were introduced in the *mst2*+ tester strain by transformation of the mutated ORF, which was marked with *URA3* from *Candida albicans*. After successful integration, *URA3* was removed by FOA counter selection. To mutate *dis32*+, the ORF was deleted with *URA3*, which was subsequently replaced by transformation of the mutated *dis32*+ ORF and FOA counter selection. The *SPBPB21E7.10-Q2Stop* early ORF truncation was generated by the insertion of an hphMX cassette.

### Plasmids

Plasmids were cloned by standard molecular biology techniques. For a list of plasmids generated in this study see S3 Table. The plasmid expressing *yth1*+ and *ura4*+ (pMB1869) was constructed by cloning *yth1*+, including 978bp upstream and 530bp downstream sequence, into KpnI/PstI digested pFY20, which was a kind gift from Mari K. Davidson. To generate the *hph*+ marked plasmids, the *ura4*+ marker of pFY20 was first replaced with *hph*+ (pMB1867).

### EMS mutagenesis, hit selection, and backcrossing

SPB2970 cells (*h- leu1-32 ura4-D18 ade6-M210 trp1*+::ade6+ nmt1+::ade6-hp+::natMX mst2Δ::kanMX*) were mutagenized as described previously [18]. Clones that grew on medium lacking adenine and lost the red color phenotype at 37˚C were backcrossed four times with the parental strains SPB2970 or SPB2971, depending on the mating type (h- or h+, respectively).

## Whole-genome sequencing

Genomic DNA was isolated from overnight cultures using the MasterPure yeast DNA isolation kit (Epicentre). Genomic DNA libraries for next-generation-sequencing were prepared from 50ng of sonicated DNA, using the NEBnext Ultra kit (NEB) following the manufacturer's protocol. Libraries were sequenced 50bp single-end on the Illumina HiSeq2500 platform. Basecalling and quality scoring was performed using RTA v1.18.64, and demultiplexing using bcl2fastq2 v2.17 (Illumina). For SNP calling we adapted a previously described pipeline [18]: For each strain, between 6.4 and 17.6 million (mean of 11.5 million) 50-nucleotide reads were generated and aligned to the Schizosaccharomyces pombe 972h- genome assembly (obtained on 17 September 2008 from http://www.broad.mit.edu/annotation/genome/schizosaccharomyces_group/MultiDownloads.html) using 'bwa' (version 0.7.15) with default parameters, but only retaining single-hit alignments ('bwa samse -n 1' and selecting alignments with 'X0:i:1'), resulting in a genome coverage between 26 and 71-fold (mean of 47-fold). The alignments were converted to BAM format, sorted and indexed using 'samtools' (version 1.3.1). Potential PCR duplicates were removed using 'MarkDuplicates' from 'Picard' (http://picard.sourceforge.net/, version 2.7.1). Sequence variants were identified using GATK (version 3.6) indel realignment and base quality score recalibration. A set of high confidence variants was identified in an initial step as known variants, followed by single nucleotide polymorphism (SNP) and INDEL discovery and genotyping for each individual strain using standard hard filtering parameters, resulting in a total of 14–103 sequence variants (mean of 68) in each strain compared to the reference genome. Finally, variants were filtered to retain only high quality single nucleotide variants (QUAL $> = 50$) of EMS type (G|C to A|T) with an allelic balance $> = 0.9$ (homozygous) that were not also identified in the parental strain (sms0), reducing the number of variants per strain to a number between 1 and 8 (mean of 3.6).

## Random mutagenesis of $yth1^+$

To generate a mutant $yth1$ plasmid library (p-$yth1^*$/$hph^+$), p-$yth1^+$/$hph^+$ (pMB1870) was transformed into $E.\ coli$ XL1-Red competent cells (Agilent). More than 200 transformed colonies were picked randomly, pooled and grown over night at 37˚C before plasmids were isolated with the PureYield Plasmid Midiprep System (Promega).

## Selection of silencing-enabling $yth1$ mutants

To rescue growth of $yth1\Delta$ $S.\ pombe$ cells, they were first transformed with an $yth1^+$ expressing plasmid (pMB1869) before the endogenous $yth1^+$ gene was deleted with a kanMX cassette, resulting in the strain SPB3646. SPB3646 was transformed with the mutant p-$yth1^*$/$hph^+$ library and grown on YES plates for 2 days at 30˚C. Cells were then replica plated on YE plates supplemented with 0.226g/l leucine, 0.226g/l lysine, 0.226g/l histidine, 0.226g/l uracil, 2g/l FOA and 100mg/l Hygromycin B (YE4S+FOA+Hygromycin B). Plasmids were recovered from yeast colonies with a silencing phenotype and subsequently sequenced by Sanger-sequencing. To confirm the silencing phenotype, the isolated plasmids were transformed in SPB3646, followed by growth on YE4S+FOA+Hygromycin B plates to force loss of the $yth1^+$ rescue plasmid and to score for colony color.

## Silencing assays

To assess $ade6^+$ expression, serial five-fold dilutions of the respective strains were plated on yeast extract (YE) plates and incubated at 30˚C for 3–4 days. Plates were stored at 4˚C overnight before pictures were taken.

To quantify initiation and maintenance rates of silencing, either single-cell-derived white (for initiation) or red (for maintenance) colonies were selected from a YE-NAT plate. A single colony was resuspended in $H_2O$ and 50–500 cells were seeded on YE plates, which were incubated at 30˚C for 4 days. Colonies were categorized and counted, after an additional overnight incubation at 4˚C, using a deep learning pipeline for high-throughput colony segmentation and classification [28].

## Chromatin immunoprecipitation (ChIP)

ChIP experiments were performed as described previously [18] with a histone H3K9me2-specific mouse monoclonal antibody from Wako (clone no. MABI0307).

## RNA isolation and cDNA synthesis

Total RNA was isolated using the MasterPure Yeast RNA Purification Kit (Epicentre). cDNA was synthesized using the PrimeScript RT Master Mix (Takara).

## Small and total RNA sequencing and analysis

Small RNA libraries were prepared with the QIAseq miRNA Library Kit (QIAGEN, Cat. No: 331505) according to the manufacturer's instructions and sequenced with an Illumina Next-Seq500 (75bp single-end). 3' adapter sequences were trimmed using cutadapt [42](version 1.18) (cutadapt -a 'adapter'—discard-untrimmed -m 18) and untrimmed or <18nt long reads were discarded. The remaining reads were aligned to the *S. pombe* genome (ASM294 version 2.24) using bowtie [43] (version 1.2.2) (bowtie -f -M 10000 -v 0 -S—best—strata). Displayed are UCSC genome browser [44] tracks of uniquely mapped reads that are normalized to one million reads (RPMs).

Total RNA libraries were prepared with TruSeq Stranded Total RNA kit (Illumina, Cat. No: 20020599) according to the manufacturer's instructions and sequenced with an Illumina HiSeq2500 (50bp single-end). RNA-seq reads were aligned to the *S. pombe* genome (ASM294 version 2.24) using STAR [45] (version 2.7.3a) (STAR—runMode alignReads—outFilterType BySJout—outFilterMultimapNmax 100—outFilterMismatchNoverLmax 0.05—outSAMmultNmax 1—outMultimapperOrder Random—outSAMtype BAM SortedByCoordinate—outSAMattributes NH HI NM MD AS nM—outSAMunmapped Within). The reads per gene were counted with featureCounts [46] of uniquely mapping reads only (useMetaFeatures = TRUE, allowMultiOverlap = FALSE, minOverlap = 5, countMultiMappingReads = FALSE, fraction = FALSE, minMQS = 255, strandSpecific = 2, nthreads = 20, verbose = FALSE, isPairedEnd = FALSE). An external feature annotation file for *S. pombe* was used based on a GFF3 file from PomBase [30] that was converted to a GTF file with rtracklayer [47]. Fragments per kilobase million (FPKMs) were calculated and averaged over three replicates. Each scatterplot depicts log2-transformed FPKM values of the wild-type against one of the mutants.

Small RNA and total RNA sequencing data have been deposited at the NCBI Gene Expression Omnibus (GEO) database and are accessible through GEO series number GSE173837.

## Quantitative real-time PCR

Real-time PCR on cDNA samples and chromatin immunoprecipitation (ChIP) DNA was performed as described using a Bio-Rad CFX96 real-time system using SsoAdvanced SYBR Green supermix (Bio-Rad) [48]. For primer sequences see S4 Table.

## Qualitative RT–PCR

PCR on cDNA was performed using the fast-cycling PCR kit (Qiagen). PCR products were analyzed by agarose gel electrophoresis. Primer sequences are listed in S4 Table.

## 3' RACE

3' RACE to assess PolyA tail length was performed as described elsewhere [49]. Briefly, first strand cDNA was synthesized from total RNA with an anchored oligo-d(T)$_{17}$VN primer using ProtScript reverse transcriptase (NEB). A first round of amplification was performed with primers mb7234 and mb13037. A second round of amplification was performed with primers mb135 and mb13038.

## Structure modelling

The structure of human CPSF-30 in complex with CPSF-160, WDR33 and PAS RNA (PDB ID 6DNH) was selected as homology model for the *S. pombe* complex based on an HHpred search [50]. Positions of *S. pombe* Yth1 point mutations, which are located in zinc finger domains 1 and 2, were mapped to HsCPSF-30 by aligning both sequences using Clustal Omega [51]. Except for *S. pombe* Yth1 phenylalanine 99 being tyrosine in human CPSF-30, respective residues were identical. *S. pombe* Cft1 and Psf2 domains (aa 1117–1349 and aa 6–404) interacting with Yth1 zinc fingers 1 and 2 (aa 48–104) share 32% (55%), 45% (64%), and 69% (88%) identities (positives) with the human homologs, respectively, based on local sequence alignments using BLAST (aligned sequence ranges: aa 1117–1349 (SpCft1), aa 6–404 (SpPsf2), aa 51–104 (SpYth1) [52]).

This suggested adequate structural conservation that allowed the analysis of mutations in *S. pombe* in the context of the human complex. A model for these point mutations was generated by mutating mapped residues in HsCPSF-30 (6DNH) using PyMOL (The PyMOL Molecular Graphics System, Version 2.0 Schrödinger, LLC).

## Supporting information

**S1 Fig. *ade6*$^+$ silencing in the newly identified point-mutants depends on a functional RNAi pathway.** Serial dilution assays showing the *ade6*$^+$ silencing phenotypes (red color when grown on YE plates) in the respective mutant cells. Silencing is not observed in the absence of Dicer (*dcr1*$^Δ$). Cells are *mst2*$^+$ but harbor mutations in individual genes as indicated. (TIF)

**S2 Fig. Most of the newly identified point-mutants with a low silencing initiation rate have a higher maintenance of silencing rate.** Comparison of initiation (A) and maintenance (B) of *ade6*$^+$ silencing in the mutant cells as indicated (*trp1*$^+$::*ade6*$^+$, *nmt1*$^+$::*ade6-hp*$^+$). Multiple individual originator colonies (white or red in A or B, respectively) were spread to single cell density on YE plates to assess initiation/maintenance of the silencing phenotype. Number of originator colonies and total counted number of colonies (in brackets) are indicated on top of the graphs. (TIF)

**S3 Fig. Expression of siRNAs in cells with weakened CPF/AAUAAA RNA interactions triggers RNAi-mediated heterochromatin silencing of protein coding genes *in trans*. (A)** RNAi-directed silencing in *yth1-R59C* cells is not unique to the *trp1*$^+$::*ade6*$^+$ silencing reporter. The *yth1-R59C* mutation was introduced into cells that express synthetic ura4-hp siRNAs instead of ade6-hp siRNAs, and a *trp1*$^+$::*ura4*$^+$ instead of a *trp1*$^+$::*ade6*$^+$ reporter. *ura4DS/E*

denotes a partial deletion of the endogenous *ura4*$^+$ gene. Silencing of the *ura4*$^+$ reporter was assessed by growth in the presence or absence of 5-FOA (which is toxic to *ura4*$^+$ expressing cells). Note that 5-FOA resistant colonies did only form in the presence of ura4-hp siRNAs and simultaneous mutation of *yth1*$^+$. **(B)** Silencing assays showing the degree of *ade6*$^+$ silencing in cells harboring mutations in *yth1*$^+$ (*yth1-R59C*) or at the 4$^{th}$ position in the PAS of the *ade6*$^+$ reporter gene. See Fig 3E for a quantification of the initiation of silencing rates in the PAS mutants. **(C)** Silencing assay demonstrating that *trp1*$^+$::*ade6*$^+$ silencing in *yth1-R59C* cells depends on RNAi (*dcr1*$^Δ$, *w/o ade6-hairpin*) and H3K9 methylation (*clr4*$^Δ$). **(D)** Silencing assay demonstrating that *trp1*$^+$::*ade6*$^+$ silencing in *pfs2-F115H* cells depends on RNAi (*dcr1*$^Δ$, *w/o ade6-hairpin*) and H3K9 methylation (*clr4*$^Δ$).
(TIF)

**S4 Fig. Impairment of the CPF phosphatase module enables RNAi-mediated heterochromatin silencing. (A)** Silencing assays demonstrating that *trp1*$^+$::*ade6*$^+$ silencing in CPF phosphatase module mutant cells (*ssu72*$^Δ$, *swd22*$^Δ$, *dis2*$^Δ$, *ppn1*$^Δ$) depends on RNAi (*dcr1*$^Δ$, *w/o ade6-hairpin*) and H3K9 methylation (*clr4*$^Δ$). **(B and C)** Comparison of initiation (B) and maintenance (C) of *ade6*$^+$ silencing in CPF phosphatase module mutant cells (*trp1*$^+$::*ade6*$^+$, *nmt1*$^+$::*ade6-hp*$^+$). Multiple individual originator colonies (white or red in B or C, respectively) were spread to single cell density on YE plates to assess initiation/maintenance of the silencing phenotype. Number of originator colonies and total counted number of colonies (in brackets) are indicated on top of the graphs.
(TIF)

**S5 Fig. Steady-state mRNA levels remain largely unaffected in cells harboring RNAi-enabling CPF mutations. (A)** Quantitative RT-PCR with primer pairs amplifying *ade6*$^+$ or *act1*$^+$ mRNAs in the respective mutant strains, which do not express any *ade6*$^+$ siRNA. mRNA levels were normalized to U6snRNA and are shown relative to the levels measured in wild-type cells. Error bars indicate standard deviation, n = 3 independent biological replicates. **(B)** Pairwise comparisons of gene expression (RNA-seq) between wild-type and CPF or *trp1*$^+$:: *ade6*$^+$ PAS mutant strains. Cells did not express primary *ade6-hairpin* siRNAs. Fragments per kilobase million (FPKMs) were calculated and averaged over three replicates. Each scatterplot depicts log2-transformed FPKM values of the wild-type against one of the mutants.
(TIF)

**S1 Table EMS type mutations identified by Illumina whole genome next-generation sequencing.**
(XLSX)

**S2 Table. List of *S. pombe* strains used in this study.**
(XLSX)

**S3 Table. List of plasmids used in this study.**
(XLSX)

**S4 Table. List of primers used in this study.**
(XLSX)

**S1 Data. Values used to generate box plot shown in Fig 2C.**
(XLSX)

**S2 Data. Values used to generate box plot shown in Fig 2D.**
(XLSX)

**S3 Data. Values used to generate graph shown in Fig 2E.**
(XLSX)

**S4 Data. Values used to generate box plot shown in Fig 3E.**
(XLSX)

**S5 Data. Values used to generate box plot shown in Fig 3G.**
(XLSX)

**S6 Data. Values used to generate graphs shown in Fig 5C.**
(XLSX)

**S7 Data. Values used to generate box plots shown in S2 Fig.**
(XLSX)

**S8 Data. Values used to generate box plots shown in S4B and S4C Fig.**
(XLSX)

**S9 Data. Values used to generate graph shown in S5A Fig.**
(XLSX)

## Acknowledgments

We thank N. Laschet for technical assistance, H. Gut for help with the design of CPF mutations, M. K. Davidson for providing the plasmid pFY20, and E. P. F. Moreno, K. Hirschfeld, and S. Smallwood for library construction and next-generation sequencing.

## Author Contributions

**Conceptualization:** Marc Bühler.

**Formal analysis:** Yukiko Shimada, Sarah H. Carl, Georg Kempf.

**Funding acquisition:** Marc Bühler.

**Investigation:** Yukiko Shimada, Merle Skribbe, Valentin Flury, Tahsin Kuzdere.

**Methodology:** Yukiko Shimada, Georg Kempf.

**Project administration:** Marc Bühler.

**Resources:** Marc Bühler.

**Supervision:** Marc Bühler.

**Validation:** Yukiko Shimada.

**Visualization:** Georg Kempf, Marc Bühler.

**Writing – original draft:** Marc Bühler.

**Writing – review & editing:** Sarah H. Carl, Merle Skribbe, Valentin Flury, Tahsin Kuzdere, Georg Kempf, Marc Bühler.

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
