## [Decision Letter · Decision Letter 0]

14 Feb 2021

Dear Dr Buehler,

Thank you very much for submitting your Research Article entitled 'An enhancer screen identifies new suppressors of small-RNA-mediated epigenetic gene silencing' to PLOS Genetics.

The manuscript was fully evaluated at the editorial level and by three independent peer reviewers, all experts in the field. The reviewers appreciated the attention to an important problem, but raised some substantial concerns about the current manuscript. Based on the reviews, we will not be able to accept this version of the manuscript, but we would be willing to review a much-revised version. We cannot, of course, promise publication at that time.

The reviewers agree that your manuscript addresses an important topic, of interest to a wide audience of researchers interested in the generation and maintenance of heterochromatic regions by siRNA. They stress the novelty of finding a role for CPF in this type of gene silencing. However, the reviewers also agree on the necessity for some additional experimentation, including providing controls that are lacking in the current version of your manuscript; please pay special attention to controls for experiments shown in Figure 1 (reviewer 1 calls this “Figure 1D” but likely means Fig. 1C). Reviewer 3 also suggests use of a second, independent reporter gene, something worth considering, as useful fission yeast strains may exist. Reviewer 2 points out two major issues, namely the lack of evidence for “unprocessed transcripts retained on chromatin” (related to Figure 4) and the absence for a phenotypic assay for phosphatase mutants, which should be included to support your conclusions. Reviewer 1 suggests ChIP assays to exclude indirect effects. Lastly, Reviewer 2 also suggests assaying ade6 transcript levels “in yth1-R59C, A4 PAS, and phosphatase mutants compared to wild-type cells”. While the reviewer admits that levels may not be drastically changed, this seems like an important issue to address in a revised version. There are several other major comments that warrant attention, including referring to previous work, as suggested by reviewers 1 and 3. Should you decide to revise the manuscript for further consideration here, your revisions should address the specific points made by each reviewer. We will also require a detailed list of your responses to the review comments and a description of the changes you have made in the manuscript.

If you decide to revise the manuscript for further consideration at PLOS Genetics, please aim to resubmit within the next 60 days, unless it will take extra time to address the concerns of the reviewers, in which case we would appreciate an expected resubmission date by email to plosgenetics@plos.org.

[LINK]

We are sorry that we cannot be more positive about your manuscript at this stage. Please do not hesitate to contact us if you have any concerns or questions.

Yours sincerely,

Michael Freitag

Associate Editor

PLOS Genetics

Wendy Bickmore

Section Editor: Epigenetics

PLOS Genetics

Reviewer's Responses to Questions

**Comments to the Authors:**

Reviewer #1: It is well known that RNAi machinery is critical for heterochromatin formation at fission yeast centromeres as siRNAs recruit CLRC (Clr4 methyltransferase complex) in cis. However, it has been reported that siRNAs cannot act in trans to trigger heterochromatin at targeted euchromatic locations with the exception of genetic backgrounds that allow such process (e.g. paf1, mst2, mlo3). In this ms. Shimada and colleagues perform a forward genetic screen that identify suppressors of the “trans-acting” small-RNA-mediated epigenetic silencing. The system, already used by the authors in previous papers, consists of a reporter strain that expresses an RNA hairpin on chromosome I that is complementary to an ade6+ reporter gene on chromosome II. In a wild type “naïve” condition, colonies are white (i.e. ade6+ ON). However, if the tester strain acquires enabling mutations yeast colonies become red (i.e. ade6+ OFF), which allows the authors to find 20 novel mutants for genes involved in RNA processing, regulation of transcription, etc. Of outstanding interest, a mutant in Yth1, a component of the conserved CPF (Cleavage and Polyadenylation Factor), display the highest levels of silencing. Such result is counterintuitive since CPF has been recently found to be involved in heterochromatin formation by an RNAi-independent mechanism. The authors also report that mutations in non-essential genes of CPF complex allow trans-silencing, proposing a model in which defective RNA processing delays nascent transcript release from chromatin, allowing sufficient time for the siRNA-guided RITS to recruit CLRC. In summary, this study is in principle interesting for the fission yeast and heterochromatin communities. However, I think that more experimental evidence is required to support the proposed model. In addition, essential controls are missing in some figures. Please find my comments and suggestions below.

Major issues:

_Figure 1D: Some critical controls are missing such as wild-type (with or without nmt1:ade6-hp+) and the identified mutants, in particular yth1-R59C, lacking the hairpin. This data must be included to rule out any indirect effect (e.g. mutants displaying red color per se). This information can be presented as dilution assays given the number of strains involved.

_Figure 2E: Can the authors rule out a mere redistribution of heterochromatin from other locations to the trp1::ade6+ region? For example, in Shelterin mutants, heterochromatin released from telomeres can relocate to centromeres thus suppressing RNAi mutants (Tadeo et al., 2013 -Jia Group)

_Figure 3E: The authors shall also show colony color pictures directly comparing yth1-R59C to PAS A4 mutants. This is a critical for ruling indirect effects caused by CPF mutants. I would expect similar levels of silencing between these mutants if the authors’ model is correct. Also, the levels of H3K9me2 in PAS A4 mutants as compared to yth1-R59C needs to be shown.

_In addition, ChIP analyses for Yth1 and Yth1-R59C is needed to rule out any indirect effect. I would expect lower enrichment for the mutant at the trp1:ade6+.

_Citations are missing as follows:

Lines 241-243: Please cite Lee et al. 2020 (PMID: 32101745), since this paper also identified CPF factors involved in heterochromatin formation.

Introduction: The section describing siRNA becoming potent mediators of RNAi-mediated epigenetic silencing in mutants such as mlo3∆ is missing an important reference. The authors must acknowledge a paper by Reyes-Turcu et al. NSMB 2011 (PMID: 21892171; Grewal group) that was first to show that cells lacking mlo3 show hairpin-induced heterochromatin assembly (see fig S7). Reyes-Turcu et al also showed that treatment of cells 6-AU enhances hairpin-induced heterochromatin assembly. The system is very similar to that used later in Kowalik et al., 2015.

The authors also discuss an RNAi positive feedback loop. This positive feedback loop was first discovered by Noma et al (PMID: 15475954) and then further described by Sugiyama et al (PMID: 15615848). These papers should be referenced along with work by the authors.

Minor comments:

_Lines 188-190: How mst2+ strains were obtained? While some information is available in Materials and Methods, it is important to point out that strains are not coming from crosses, so it is not mere maintenance of silencing state carried from the mst2∆ background.

_Fig. 2C. Are the other mutants affecting initiation of silencing as well? Any additional data would be useful in a supp. figure. Similarly, initiation of silencing rates in CPF mutants would be helpful as well.

_Fig. 2E. p values should be written as “0.01” instead of “0,01” and so on.

_Fig. 3A: I wonder how CPF mutants were obtained. Where these strains obtained from crosses involving an already ade6+ silenced? Please clarify.

_Fig. 4B-D. If data is available for pfs2 mutant, that should be included.

_Fig. 4C. Is the reverse oligo annealing after the cleavage site? Please clarify.

_Table S3 and S4: please indicate in which figure/s the strains or oligos have been employed. That will help the reader.

Reviewer #2: The role of small interfering non-coding RNAs in promoting de novo heterochromatin formation in cis was described almost two decades ago using the fission yeast Schizosaccharomyces pombe. Since then, many groups have dissected in detail many of the different components that are required for establishing RNAi-dependent heterochromatin domains at non-coding repeats.

Interestingly, contrary to the effect observed in cis, it has consistently been reported that in wild-type cells, siRNAs derived from a hairpin construct (or alternatively from a source gene inserted within heterochromatin repeats) do not lead to effective transcriptional silencing of a complementary target sequence in trans. However, this refractory effect was shown to be suppressed in the presence of specific enabling mutations, such as those in the Paf1 complex or the mRNA export factor Mlo3.

In this manuscript, Shimada et al. report the identification of 20 novel mutant alleles that enable hairpin-derived siRNAs to mediate transcriptional silencing of a target ade6 gene in trans. The authors followed the same strategy that led to the identification of Paf1C mutants in a previous study (Kowalik et al., 2015) but here, to maximise the chances of obtaining trans silencing-enabling mutations, the authors performed their EMS screen in a mst2D background.

Of 20 identified alleles, the authors decided to focus on yth1-R59C, encoding a mutant of the CPF complex subunit Yth1, which showed the strongest silencing phenotype. Indeed, yth1-R59C cells exhibited a remarkably high rate of silencing establishment. Furthermore, using a mutagenized plasmid library-based strategy, the authors identified additional mutations in yth1 that enable siRNA-mediated silencing. Modelling the identified yth1 mutations on the structure of human CPF revealed that R59C has the potential to disturb the interaction with adenosine at position 4 of the pre-mRNA PAS. Consistent with this hypothesis, the authors showed that strains harbouring mutations in A4 of the ade6 PAS are able to establish siRNA-mediated silencing in the absence of yth1-R59C or other enabling mutations.

A mutation in Pfs2 (another CPF component) was also found to allow siRNA-mediated silencing to occur in trans, and a similar effect was proposed to be achieved upon deletion of several CPF phosphatase module components.

Lastly, in an attempt to consolidate their previously-proposed kinetic nascent transcript processing model for the inhibition of siRNA-mediated de novo heterochromatin establishment, the authors show that in strains harbouring yth1-R59C or A4 mutations, ade6 pre-mRNA cleavage is reduced.

The manuscript is in general well-written and easy to follow. Even though mutations that enable siRNA-mediated silencing in trans have been described before, this work is of interest as it certainly provides a list of novel enabling mutations, some of which occurred in uncharacterized factors or components with biological roles not previously associated with siRNA-mediated silencing. However, further mechanistic insights should be provided regarding the role of yth1 mutations in enabling siRNA-mediated silencing through the previously proposed kinetic nascent transcript processing model, as the presented experiments only further solidify the correlation between siRNA-mediated silencing establishment and levels of non-cleaved pre-mRNA.

Major points:

1. In the abstract, the authors state: ‘Our results provide direct evidence for a kinetic nascent transcript processing model‘, yet experiments presented in Figure 4 only show that yth1-R59C cells exhibit reduced cleavage of ade6 pre-mRNA. No evidence of unprocessed transcripts being retained on chromatin is provided. The authors should directly test if an increase in chromatin-bound unprocessed ade6 transcripts in yth1-R59C cells is associated with the observed high silencing establishment rate. Can the high silencing establishment frequency of yth1-R59C be suppressed by engineering pre-mRNA cleavage (e.g. ribozyme insertion as used in S. cerevisiae Mol Cell 2003 12:711 ) and release from chromatin?

2. Although the establishment of siRNA-mediated silencing at a complementary sequence in trans has previously been shown to correlate with secondary siRNA generation at the target site, no experiments aimed at detecting those were performed here. Confirming the presence of secondary siRNAs is of special importance in yth1-R59C cells, since the authors performed substantial follow-up experiments to elucidate the mechanism by which this mutation enables silencing.

3. Line 303-305: ‘CPF mutants that we have investigated in this study phenocopy wild type cells, i.e. neither growth nor gene expression is noticeably affected.’

- There seems to be a small but noticeable difference in colony size when comparing yth1 mutants to wild-type cells. These differences can be observed in YES plates of Figure 3B and Figure 4A (also for dis2D and ssu72D mutants).

- Quantitative analyses of ade6 mRNA levels (without ade6-hp) in yth1-R59C, A4 PAS, and phosphatase mutants compared to wild-type cells are not shown in this manuscript. Although a strong change in ade6 mRNA levels is not expected in yth1-R59C and A4 mutants by looking at the phenotypes shown in Figure 2B (no ade6-hp, dcr1+) and 3E, ade6 mRNA levels should be quantitatively assessed in these conditions and compared to levels observed in wild-type cells. Figure 4B shows differences in cleavage among mutants but does not compare total levels. Figure 4C (bottom) suggests that ade6 mRNA levels in mutants are equal to wild-type but a loading control was not included in this experiment.

- For phosphatase mutants, in addition to quantitatively assessing ade6 mRNA levels, a phenotype (colour) assay should be shown in the absence of ade6-hp or alternatively in the presence of ade6-hp but in a dcr1D background. Data shown in Figure 4 are not enough to support the conclusion that these mutations enable siRNA-mediated silencing (this is of importance given that these mutations did not come up in the screen which excluded mutants in the adenine pathway).

Other points:

- Figure 4 and text lines 331/337 refer to yth1-G59C instead of yth1-R59G which made the respective figure and text highly confusing.

- Some pink colonies are formed in p-yth1+-containing cells in YE medium. Is this due to differences in plasmid stability among clones?

- Given that phosphatase mutants do not affect pre-mRNA cleavage, could the authors speculate on how these mutations lead to siRNA-mediated silencing?

- The use of the term ‘tester strain’ is at times confusing as it is unclear whether this refers to the original mst2+ tester strain used in Kowalik et al. (2015) or the mst2D strain used in this study. I suggest adding mst2+ or mst2D in between brackets every time ‘tester strain’ is mentioned.

- Figure 2C/D/E. Please indicate genotype either in panels or in legend (no mention of ade6-hairpin is made for those panels).

- Including the silencing establishment frequency for each of the identified mutant alleles (both in mst2D and mst2+ backgrounds) in Table 2 would be helpful to the community.

- Line 291 - Forgoing > Foregoing?

- Figure 4C/D: Diagrams illustrating primer locations (C (top) and D) are confusing at they seem to indicate that the amplicon in D is longer than C (top).

- Although a reference is provided, given the importance of the procedure for this study, it would be appropriate to fully describe the computational pipeline followed to identify SNPs, especially as it is indicated that an ‘adapted’ pipeline was used.

Reviewer #3: Heterochromatin is essential for genome stability and gene expression. Small interference RNAs (siRNAs) have been shown to plays an important role in peri-centromeric heterochromatin, which contains repetitive DNA, in fission yeast. However, the protein-coding genes in euchromatin are resistant to siRNA-mediated heterochromatin formation. The mechanism used for counteracting heterochromatin assembly remains largely unclear. Through an elegant genetic screen, Shimada et al. identified more than twenty mutant alleles that enable de novo formation of heterochromatin at a protein-coding gene in euchromatin. By further characterizing a novel factor they identified, the pre-mRNA cleavage factor Yth1, they provides new insight into how small-RNA-directed de novo formation of heterochromatin in euchromatin is inhibited. This work is scientifically interesting and significant. The experiments were in general well designed and carefully executed, and most of the results are convincing. There are several concerns as listed below.

Major:

1. There is a concern about the indirect effect on the expression of the ade6 reporter used in the genetic screen in the mutant backgrounds. The study will be strengthened by confirming the de novo heterochromatin formation using growth assays having a different reporter.

2. Figure 1C should have a wild type control.

3. The study would be benefited by analyzing mutants with de novo heterochromatin formation in the clr4 mutant background using same growth assays. This would further confirm that the de novo heterochromatin formation depends on H3K9 methylation.

4. It would be interesting to compare the de novo heterochromatin silencing in these mutants with the native pericentromere silencing, such as H3K9 methyation and RNA transcription level of the reporter in the two regions.

5. In Introduction, Line 94, the statement “Whereas siRNAs originating from heterochromatic repeats function well in cis to sustain H3K9 methylation, they do not act in trans to mediate de novo formation of heterochromatin at complementary sequences in wild type cells “ is inaccurate. It has been shown that siRNAs can act in trans to induce heterochromatin formation at complementary sequences in pericentromeres in Chromosome III in wild type fission yeast cells (Haijin He, et al. Cell Reports, 2016).

Minor

1. The media used in this paper should be listed in the Materials and Methods section.

2.Line 570, it should be “wild-type”, not “wildtype”.

**Have all data underlying the figures and results presented in the manuscript been provided?**

Reviewer #1: Yes

Reviewer #2: **No: **The source data underlying Figure 2C/D/E, Figure 3E/G and Figure 4B do not seem to be provided.

Reviewer #3: Yes

PLOS authors have the option to publish the peer review history of their article (what does this mean?). If published, this will include your full peer review and any attached files.

Reviewer #1: No

Reviewer #2: No

Reviewer #3: No

---

## [Editor Report · Decision Letter 1]

4 Jun 2021

Dear Dr Bühler,

We are pleased to inform you that your manuscript entitled "An enhancer screen identifies new suppressors of small-RNA-mediated epigenetic gene silencing" has been editorially accepted for publication in PLOS Genetics. Congratulations!

In your point-for-point response you addressed all comments of the three reviewers, you added the requested experimental data, and you made changes that were suggested to soften some of your conclusions, reflecting the data produced in your study. Thanks for sending this interesting work to PLOS Genetics! 

Yours sincerely,

Michael Freitag

Associate Editor

PLOS Genetics

Wendy Bickmore

Section Editor: Epigenetics

PLOS Genetics

Comments from the reviewers (if applicable):

**Data Deposition**

http://datadryad.org/submit?journalID=pgenetics&manu=PGENETICS-D-20-01839R1

**Press Queries**

---

## [Editor Report · Acceptance letter]

16 Jun 2021

PGENETICS-D-20-01839R1 

An enhancer screen identifies new suppressors of small-RNA-mediated epigenetic gene silencing 

Dear Dr Bühler, 

We are pleased to inform you that your manuscript entitled "An enhancer screen identifies new suppressors of small-RNA-mediated epigenetic gene silencing" has been formally accepted for publication in PLOS Genetics! Your manuscript is now with our production department and you will be notified of the publication date in due course.

With kind regards,

Zsofi Zombor

PLOS Genetics

On behalf of:
